# Analysis of clinical trial registry entry histories using the novel R package cthist

**Benjamin Gregory Carlisle**⊕*

Berlin Institute of Health at Charité –Universitätsmedizin Berlin, QUEST Center for Responsible Research, Berlin, Germany

* benjamin.carlisle@bih-charite.de

## Abstract

Historical clinical trial registry data can only be retrieved by manually accessing individual clinical trials through registry websites. This limits the feasibility, accuracy and reproducibility of certain kinds of research on clinical trial activity and presents challenges to the transparency of the enterprise of human research. This paper presents `cthist`, a novel, free and open source *R* package that enables automated scraping of clinical trial registry entry histories and returns structured data for analysis. Documentation of the implementation of the package `cthist` is provided, as well as 3 brief case studies with example code.

## Introduction

Prospective registration of clinical trials in a public database such as ClinicalTrials.gov or the German Clinical Trials Register, DRKS.de is ethically required of investigators by the Declaration of Helsinki [1], a prerequisite for publication according to the ICMJE [2], and for certain clinical trials, it is mandated by law [3]. A rationale for preregistration of a clinical trial is that it helps to reduce certain biases in the medical literature, such as selective reporting and nonpublication bias, however preregistration plays many other roles within the enterprise of clinical research. Reviewers of clinical trial journal publications can use registry entries to ensure that the published record corresponds to the trial that was planned. Clinical trial registries are also used as tools to help enrol prospective patients, providing them with information about new and ongoing studies by location and disease area. Systematic reviewers make use of clinical trial registries to synthesize available clinical evidence. Meta-researchers analyze data from trial registries to describe programmes of human research, evaluate them, or hold clinical investigators accountable to common standards of good research practice.

Clinical trial registry entries are often regarded as an immutable record of a trial's registration, however they can be modified by the responsible party at any time, and changes to clinical trial registry entries between initial registration and last registration are common [4]. Researchers who analyze clinical trial registry data but fail to account for potential changes to registry entries run the risk of making serious methodological errors, such as failure to account for variable follow-up time to an event of interest [5]. Certain research questions and insights into the enterprise of human research are not possible or not feasible to address without a means of accessing and analyzing the history of changes for an entire cohort of clinical trials in

**Competing interests:** The authors have declared that no competing interests exist.

a systematic way. For example, analyzing the rates at which clinical trials achieve their originally anticipated enrolment goals would be very difficult without a method for accessing the original clinical trial registry record and the record that was active at the time of completion.

Prior to the publication of this package, options for accessing historical trial registry data were limited. While ClinicalTrials.gov does provide an Application Programming Interface (API) that allows access to the most recent version of clinical trial registry entries, the API does not allow access to historical clinical trial registry data, and DRKS.de does not provide an API at all. There are other 3rd-party options to access clinical trial data from DRKS [6], however these do not provide historical data either. Hence, the most straightforward way to access historical versions of a trial registry entry was by manually visiting the website for the clinical trial registry and recording the trial data systematically in a spreadsheet. While this was often feasible when considering a single clinical trial's history, this method applied to a large cohort of clinical trials is extremely labour-intensive, error-prone and limits reproducibility. The challenge this presents to the feasibility of certain kinds of meta-research has been remarked on elsewhere. (See Al-Durra et al 2020 [7], supplementary appendix L.).

Alternatively, a researcher could download the entire database of a clinical trial registry at regular intervals; however this is extremely resource-intensive, both in terms of data storage and the time required to process the data, which also limits the reproducibility of these methods.

In order to provide a tool that makes clinical trial registry history research accessible, feasible and reproducible for patients, reviewers, meta-researchers and systematic reviewers, `cthist`, the *R* package presented here, was developed. This package provides functions that allow access to historical clinical trial registry data from ClinicalTrials.gov and DRKS.de without bringing about the human error that would result from manual searches and extraction, or the resources that would be required for regularly mass-downloading and processing of the entire registry database. In what follows, the implementation of `cthist` is presented and its use is described with 3 potential case studies with example code.

## Methods

### Availability and requirements

The *R* package `cthist` can be downloaded from CRAN [8], or the development version can be installed from GitHub (https://github.com/bgcarlisle/cthist). Internal package documentation provides arguments and examples for the included functions.

The package was written for *R* [9], and depends on the following *R* packages: `dplyr` [10], `httr` [11], `jsonlite` [12], `magrittr` [13], `readr` [14], `rlang` [15], `rvest` [16], `selectr` [17], `stringr` [18], `tibble` [19], `polite` [20].

### Package implementation

The `cthist` package provides functions for downloading historical clinical trial registry data for trials registered on ClinicalTrials.gov and trials registered on DRKS.de. Future versions may include functions to download historical clinical trial registry data from other registries.

The functions for downloading historical trial data have been implemented for both ClinicalTrials.gov and DRKS.de. As much as possible given the constraints of the source database, the output returned by the function for ClinicalTrials.gov is similar to the output returned by the corresponding function for DRKS.de.

Six functions are provided by `cthist`: `clinicaltrials_gov_dates()`, `drks_de_dates()`, `clinicaltrials_gov_version()`, `drks_de_version()`, `clinicaltrials_gov_download()` and `drks_de_download()`. Three allow

downloading of data from ClinicalTrials.gov, and the other three are their counterparts for DRKS.de. These are described in detail below.

**clinicaltrials_gov_dates() and drks_de_dates().** The functions `clinicaltrials_gov_dates()` and `drks_de_dates()` take a trial registration number (NCT number or DRKS id, respectively) as an argument and provide a character vector of ISO-8601 formatted dates, one for each update to the registry entry history.

For ClinicalTrials.gov, these dates are retrieved by downloading the HTML for the index of history changes, selecting all the cells in the "Submitted Date" column from the table of study record versions, extracting the text using the `polite` *R* package, and re-formatting the dates.

Similarly for DRKS.de, dates are retrieved by downloading the HTML for the change history page, selecting all the cells in the "Date" column from the published versions table, extracting the text using the `polite` *R* package, and reformatting the dates.

These functions then return a character vector containing dates in the case of success, and in the case of an error (e.g. inability to connect to the internet), they return the word "Error" and print out an explanatory error message in the *R* console.

**clinicaltrials_gov_version().** The function `clinicaltrials_gov_version()` downloads clinical trial data for the NCT number and version number (starting with 1 as the earliest version) indicated by the function's arguments. The HTML for the historical version page is downloaded using `polite` and individual data points are extracted using a combination of cascading style sheet (CSS) selectors and regular expressions (regex). A list of 16 data points are returned by this function: overall status, enrolment, start date, start date precision, primary completion date, primary completion date precision, primary completion date type, minimum age, maximum age, sex, gender based, accepts healthy volunteers, inclusion criteria, outcome measures, contacts and sponsors.

See Table 1 for a list of the extracted data points, the CSS selectors that identify the HTML elements in which they are found, and the regular expressions used to extract them. For example, the overall status for the version of the trial in question is drawn from cells in the table with CSS id attribute `#StudyStatusBody` that have text that matches the regular expression `Overall Status: ([A-Za-z, ]+)`. Similarly, data points are extracted using CSS and regex as reported in Table 1 for enrolment, start date, and primary completion date.

The text for start date and primary completion date are transformed from their original format (Month Date, Year) to ISO-8601 (YYYY-MM-DD). In the case that a date is given only as accurately as the month, it would be rounded to the beginning of the month (e.g. "September 2009" would be rounded to 2009-09-01) and the `start_date_precision` or `primary_completion_date_precision` column would indicate that it has been rounded to the month, otherwise the relevant column will report that it is accurate to the day.

The minimum age, maximum age, sex, gender based, and accepts healthy volunteers data points are extracted from the HTML using the CSS selectors and regular expressions indicated in Table 1.

Inclusion criteria are extracted as the contents of the cell of the table with CSS id attribute `#EligibilityBody` that immediately follows the cell that contains the label "Criteria:". The lines of text comprising the contents of this cell are encoded as JavaScript Object Notation (JSON) to preserve the data structure.

Outcome measures are extracted from rows of the table with CSS id attribute `#ProtocolOutcomeMeasuresBody`. Each row in the original HTML table that contains an outcome measure is copied to a data frame with three columns: `section`, `label` and `content`. Because section headings are encoded in the original HTML as table rows where the second cell contains no text, the `section` is the text of the most recently preceding row where the

**Table 1. Cascading Stylesheet (CSS) selectors and regular expressions indicating HTML elements and the text to be extracted from them on ClinicalTrials.gov and DRKS.de by `clinical-trials_gov_version()` and `drks_de_version()`, respectively.**

| | ClinicalTrials.gov | | DRKS.de | |
| --- | --- | --- | --- | --- |
| | CSS | Regular expression | CSS | Regular expression |
| Overall status | `#StudyStatusBody` | `Overall Status: ([A-Za-z, ]+)` | - | - |
| Recruitment status | - | - | `li.state` | `Recruitment Status: ([A-Za-z, -]+)` |
| Enrolment | `#StudyDesignBody` | `Enrollment: ([A-Za-z0-9 \\[\\]]+)` | `li.targetSize` | `[0-9]+` |
| Enrolment type | - | - | `li.targetSize` | `Planned/Actual: ([A-Za-z]+)` |
| Start date | `#StudyStatusBody` | `Study Start: ([A-Za-z0-9, ]+)` | `li.schedule` | `[0-9]{4}/[0-9]{2}/[0-9]{2}` |
| Primary completion date | `#StudyStatusBody` | `Primary Completion: ([A-Za-z0-9, \\[\\]]+)` | - | - |
| Primary completion date type | `#StudyStatusBody` | `(\\[[A-Za-z]+\\])` | - | - |
| Closing date | - | - | `li.deadline` | `[0-9]{4}/[0-9]{2}/[0-9]{2}` |
| Minimum age | `#EligibilityBody` | `Minimum Age: ([0-9]+) Years` | `li.minAge` | `Minimum Age: ([A-Za-z0-9 ]+)` |
| Maximum age | `#EligibilityBody` | `Maximum Age: ([0-9]+) Years` | `li.maxAge` | `Maximum Age: ([A-Za-z0-9 ]+)` |
| Sex | `#EligibilityBody` | `Sex: ([A-Za-z]+)` | - | - |
| Gender | - | - | `li.gender` | `Gender: ([A-Za-z ]+)` |
| Gender based | `#EligibilityBody` | `Gender Based: ([A-Za-z]+)` | - | - |
| Accepts healthy volunteers | `#EligibilityBody` | `Accepts Healthy Volunteers: ([A-Za-z]+)` | - | - |
| Inclusion criteria | `#EligibilityBody` | ** | - | - |
| Additional inclusion criteria | - | - | `.inclusionAdd` | ** |
| Exclusion criteria | - | - | `.exclusion` | ** |
| Primary outcomes | - | - | `p.primaryEndpoint` | ** |
| Secondary outcomes | - | - | `p.secondaryEndpoints` | ** |
| Outcome measures | `#ProtocolOutcomeMeasuresBody` | ** | - | - |
| Contacts | `#ContactsLocationsBody` | ** | `ul.addresses li.address` | ** |
| Sponsors | `#SponsorCollaboratorsBody` | ** | - | - |

Asterisks (**) indicate table data parsed and encoded as JSON rather than extracted using simple regular expressions. A single hyphen (-) indicates that this data point is not available to be downloaded for this clinical trial registry.

second cell is empty. The `label` is the text in the first cell of the row. The `content` is the text contained in the second cell of the row. The data frame is encoded as JSON to preserve the data structure.

Contact information is extracted from rows of the table with CSS `id` attribute `#ContactsLocationsBody` that come before a row labelled "Locations:". Each row in the original HTML table that contains contact information is copied to a data frame with two columns: `label` and `content`. The `label` is the text in the first cell of the row. The `content` is the text contained in the second cell of the row. This data frame is encoded as JSON to preserve the data structure.

Sponsors and collaborators are extracted from rows of the table with CSS `id` attribute `#SponsorCollaboratorsBody`. Each row in the original HTML table that contains sponsor or collaborator information is copied to a data frame with two columns: `label` and `content`. The `label` is the text of the first cell of the row, and the `content` is the text of the second cell. This data frame is encoded as JSON to preserve the data structure.

For all columns that are encoded as JSON, these can be converted back to data frames using the `fromJSON()` function implemented by the *R* package `jsonlite`.

Upon successful download and parsing of a clinical trial registry history entry, the function returns a named list of these 16 data points. In case of error, the text "Error" is returned so that `clinicaltrials_gov_download()`, which calls this function, can continue in the case of an error in downloading, while still indicating which rows need to be downloaded again, to aid in downloading large data sets.

**drks_de_version().** The function `drks_de_version()` downloads clinical trial data for the DRKS number and version number (starting with 1 as the earliest version) indicated by the function's arguments. The HTML for the historical version page is downloaded using `polite` and individual data points are extracted using a combination of cascading style sheet (CSS) selectors and regular expressions (regex). A list of 13 data points are returned by this function: recruitment status, start date, closing date, enrolment, enrolment type, minimum age, maximum age, gender, additional inclusion criteria, exclusion criteria, primary outcomes, secondary outcomes and contacts. Because the data provided in the DRKS.de historical version page do not have a one-to-one correspondence with their counterparts on ClinicalTrials.gov, the columns extracted from the two registries differ.

See Table 1 for a list of the extracted data points, the CSS selectors that identify the HTML elements in which they are found, and the regular expressions used to extract them. For example, the recruitment status for the version of the trial in question is drawn from the bullet point with CSS selector `li.state`, taking the text that matches the regular expression `Recruitment Status: ([A-Za-z, -]+)`. Similarly, data points are also extracted using CSS and regex as reported in Table 1 for start date, closing date, enrolment, enrolment type, minimum age, maximum age, and gender.

Additional inclusion criteria, exclusion criteria, primary outcomes and secondary outcomes are taken from HTML elements with CSS selectors `.inclusionAdd`, `.exclusion`, `p.primaryEndpoint` and `p.secondaryEndpoints` respectively, encoded as JSON to preserve the data structure.

Contact information is returned as a data frame with columns `label`, `affiliation`, `telephone`, `fax`, `email`, `url`. Each row in this data frame is populated by one bullet with CSS selector `ul.addresses li.address` from the original HTML. The `label` column is extracted from the HTML `<label>` node. The `affiliation` column is extracted from the bullet with CSS selector `li.address-affiliation`. The `telephone`, `fax`, `email` and `url` columns are extracted from the bullets with CSS selector `.address-telephone`, `.address-fax`, `.address-email` and `.address-url`, respectively

excluding the label text for each. The data frame is encoded as JSON to preserve the data structure.

Upon successful download and parsing of a clinical trial registry history entry, the function returns a list of these 13 data points. In case of error, the text "Error" is returned so that `drks_de_download()`, which calls this function, can continue in the case of an error in downloading, while still indicating which rows need to be downloaded again, to aid in downloading large data sets.

**clinicaltrials_gov_download() and drks_de_download().** The functions `clinicaltrials_gov_download()` and `drks_de_download()` loop through the trial registry numbers provided to them in the first argument (NCT numbers or DRKS id's, respectively), and for each one, download a list of all the dates on which the trial registry entry was updated, using `clinicaltrials_gov_dates()` or `drks_de_dates()`, respectively (described above). For each version of each trial, the functions download the clinical trial registry entry version using `clinicaltrials_gov_version()` or `drks_de_version ()`, respectively. Each downloaded registry entry historical version is written to the filename specified in the second argument to the function, formatted as a CSV.

If no filename is specified, the function will return a data frame containing one historical version of a clinical trial per row.

In the case of connexion failure or server error when downloading a version of a trial record, only the text "Error" as described above, will be reported in the overall status or recruitment status column for that version. If a filename is specified, the function will return TRUE in the case that all rows were downloaded without reporting an error and FALSE in the case that an error was detected. A FALSE return alerts the user to re-run the function with the same arguments in order to remove the rows that contain errors and re-download them. On running the function again with the same arguments, it removes rows that have been marked "Error" and tries to download them again.

## Results

The following is a description of 3 case studies of research questions regarding clinical trial registry entries that are difficult or un-feasible to answer without a tool for mass-downloading clinical trial registry history data. Case studies 1 and 3 provide example code for analyzing clinical trials from ClinicalTrials.gov and example 2 provides example code for DRKS.de, however any of the three could be re-written easily for the other clinical trial registry with minor changes.

### Case study 1: Assessing change in length to recruitment period

Changes in recruitment length to a clinical trial have been used as one part of a measure of the feasibility of clinical trials [21]. In order to evaluate changes to recruitment period lengths for a cohort of clinical trials, it is necessary to mass-download enrolment data from historical versions of ClinicalTrials.gov records.

As described in R Code Box 1, clinical trials meeting the study's inclusion criteria were downloaded by performing a search via the web front-end of ClinicalTrials.gov. Search results were saved as a comma-separated value (CSV) file named `SearchResults.csv`. The `NCT Number` column in this file is parsed and used as an argument for the `clinicaltrials_-gov_download()` function implemented in package `cthist`, which then downloads the records for all the clinical trial registry entries in this sample and saves them as `historical_versions_1.csv`. This script parses these results and the percentage change of the

```
## R Code Box 1

library(tidyverse)
library(cthist)
library(lubridate)

## To reproduce these methods, conduct a search on the
## ClinicalTrials.gov web front-end and download the result as a
## comma-separated value (CSV) file named `SearchResults.csv`. Save
## this file to your R working directory. This file provides `cthist`
## with the list of NCT numbers for which to download historical trial
## registry data.

## Read the downloaded CSV into memory, and select only the `NCT
## Number` column
trials <- readr::read_csv("SearchResults.csv") %>%
    select(`NCT Number`) %>%
    pull(`NCT Number`)

## Download historical clinical trial data for all NCT numbers in the
## CSV downloaded from ClinicalTrials.gov
clinicaltrials_gov_download(trials, "historical_versions_1.csv")

## Read historical versions from the data in the CSV generated above
hv <- read_csv("historical_versions_1.csv")

## Define follow-up time as 1 year
followup <- lubridate::years(1)

## We will define the start date at launch and completion date at
## launch to be the start date and completion date reported on the
## first version of the trial where that version's overall status is
## listed as "Recruiting"

## To obtain these dates, we will consider only versions with an
## overall status of "Recruiting" and a non-NA start and completion
## date; we will filter for the first row (corresponding to the
## earliest version), and select only the NCT number and the start and
## completion dates, which we will rename `launch_start_date` and
## `launch_completion_date`
launch_dates <- hv %>%
    filter(
        overall_status == "Recruiting" &
        ! is.na(study_start_date) &
        ! is.na(primary_completion_date)
    ) %>%
    group_by(nctid) %>%
    slice_head() %>%
```

```
        ungroup() %>%
        select(nctid, study_start_date, primary_completion_date) %>%
        rename(launch_start_date = study_start_date) %>%
        rename(launch_completion_date = primary_completion_date)

## Join the original start dates to every row of each trial and remove
## trials where there was no historical version posted after the trial
## started
hv <- hv %>%
        left_join(launch_dates, by = "nctid") %>%
        filter(! is.na(launch_start_date))

## We will define the start date at follow-up and the completion date
## at follow-up to be the start date and completion date reported on
## the version of the trial that was active at the number of days
## after the start date at launch specified in the `followup` variable

## To obtain these dates, we will consider only versions with a date
## that is less than 1 year after the launch start date and where the
## start and completion dates are not NA; we will filter for the last
## row (corresponding to the latest version), and select only the NCT
## number and the start and completion dates, which we will rename
## `start_date_fup` and `completion_date_fup`
dates_fup <- hv %>%
        filter(
            version_date <= launch_start_date + followup &
            ! is.na(study_start_date) &
            ! is.na(primary_completion_date)
        ) %>%
        group_by(nctid) %>%
        slice_tail() %>%
        ungroup() %>%
        select(nctid, study_start_date, primary_completion_date) %>%
        rename(start_date_fup = study_start_date) %>%
        rename(completion_date_fup = primary_completion_date)

## Join the start and end dates as reported at launch and at follow-up
trial_dates <- launch_dates %>%
        left_join(dates_fup) %>%
        mutate(
            recruitment_length_at_launch =
                launch_completion_date -
                launch_start_date
        ) %>%
        mutate(
            recruitment_length_at_fup =
                completion_date_fup -
                start_date_fup
```

```
    ) %>%
    mutate(
        recruitment_length_change =
            paste0(
                round(
                    100 * as.numeric(recruitment_length_at_fup) /
                        as.numeric(recruitment_length_at_launch) - 100,
                    digits=0
                ),
                "%"
            )
    ) %>%
    select(
        nctid,
        recruitment_length_at_launch,
        recruitment_length_at_fup,
        recruitment_length_change
    )

## Write result as a CSV to disk
trial_dates %>%
    write_csv("trial_dates.csv")
```

length of the recruitment period between the first registry entry version posted with an overall status of "Recruiting" and the registry entry version that was active 1 year later.

Nearly identical methods to the above were applied to a cohort of SARS-CoV-2 treatment and prevention efficacy trials that were initiated between 2020-01-01 and 2020-06-30, downloading historical clinical trial versions using `cthist` [21]. One goal of this study was to assess the feasibility of clinical trials where part of the definition of feasibility was that an ongoing trial may be unfeasible if it is ongoing, but its recruitment period has been extended to at least twice as long as the original anticipated length in the version of the ClinicalTrials.gov record at the time of trial start. Assessing a large cohort of clinical trials for changes to recruitment period length would have been labour-intensive, error-prone and difficult to reproduce without the use of an automated tool for downloading clinical trial registry histories.

## Case study 2: Identifying changes to outcome measures

Outcome switching in clinical trials is a common practice [22] in which the outcomes that were pre-specified in a clinical trial registry differ from those that are published in the corresponding journal publication. Unreported outcome switching may mislead readers or introduce bias [23]. Because a clinical trial registry entry may be updated at any time, it is necessary to consult not just the most recent version of a clinical trial registry entry, but to review all the versions in the trial registry history in order to determine whether, when, and in what manner they were changed.

The code presented in R Code Box 2 will take a list of DRKS id's from a CSV downloaded from the web front-end of DRKS.de, download all the historical versions of those trials and

```
## R Code Box 2

library(tidyverse)
library(cthist)

## To reproduce these methods, conduct a search on the DRKS.de web
## front-end and download the result as a comma-separated value (CSV)
## file. The CSV download option on DRKS.de produces a zipped
## semicolon-delimited data file, which must be unzipped and saved to
## your R working directory as `trials.csv` before reading. This file
## provides `cthist` with the list of DRKS id's for which to download
## historical trial registry data.
trials <- readr::read_delim("trials.csv", ";") %>%
    select(`drksId`) %>%
    pull(`drksId`)

## Download historical clinical trial data for all DRKS id's in the
## CSV downloaded from DRKS.de
drks_de_download(trials, "historical_versions_2.csv")

## Read historical versions from the data in the CSV generated above
hv <- read_csv("historical_versions_2.csv")

## Identify "run lengths" (the number of rows that contain specified
## columns of equal value) for outcomes measures within a trial; This
## "run lengths" object will allow us to number each "run" of outcomes
outcome_runs <- rle(
    paste(hv$drksid, hv$primary_outcomes, hv$secondary_outcomes)
)

## Make an `outcome_run` column that assigns a number to each "run" of
## identical outcomes in the original historical versions data frame
hv <- hv %>%
    mutate(
        outcome_run = rep(
            seq_along(outcome_runs$lengths),
            outcome_runs$lengths
        )
    )

## Group by "runs" of outcomes and select only the first of each. This
## will produce a data frame with one row per "run", indexed by the
## NCT number and the date on which the outcome measures changed
outcome_changes <- hv %>%
    group_by(outcome_run) %>%
    slice_head() %>%
    ungroup() %>%
    select(drksid, version_date, primary_outcomes, secondary_outcomes)
```

```
## Write result as a CSV to disk
outcome_changes %>%
    write_csv("outcome_changes.csv")
```

determine which updates to each clinical trial represents a change in the trial's outcome measures. This script will identify all changes, from ones as major as the malicious switching of primary and secondary outcomes, to ones as minor as the correction of typos, or even a single-character white-space change. The work of determining whether the change should be mentioned in a final journal publication remains to be done by human curation of the script's output data.

These methods have been applied to an ongoing study to identify changes to a clinical trial's outcomes as reported on ClinicalTrials.gov or DRKS.de at key time points in the course of each trial and after its completion [24]. Changes between versions of a registry entry that are identified automatically by cthist will be assessed manually by human raters to determine whether this represents a meaningful change to the outcomes, such as an added or modified outcome measure, and if so what kind, or a minor cosmetic change (e.g. correcting a typo). While the methods for this project are not fully automated, it is only feasible to do because of the use of cthist, which identifies the trials and even the versions that need to be scrutinized by human raters. It would have been impractical to manually access the clinical trial registry histories for all 1897 trials in the study sample and check for changes in outcome measures by hand.

### Case study 3: Correcting for variable follow-up time

Let us consider the hypothetical case of a meta-researcher who wishes to characterize phase 3 glioblastoma clinical trial activity in terms of how many clinical trials are stopped (overall status changed to "Terminated", "Suspended" or "Withdrawn"). It may be tempting to search on the ClinicalTrials.gov web front-end for all or phase 3 trials with an indication of "glioblastoma" whose overall status is "Terminated", "Suspended" or "Withdrawn", count the results and report them as a fraction of the number of results for the same search without the overall status restriction. As of the writing of this manuscript, 17.2% phase 3 glioblastoma trials on ClinicalTrials.gov (16 out of 93 on 2022-01-05) had an overall status of "Terminated", "Suspended" or "Withdrawn".

This strategy does not account for variable follow-up time among the clinical trials in the sample. A trial that was registered yesterday, for example, may yet go on to be withdrawn, if it were given the same follow-up time as the trials in the sample that were registered five years ago. By failing to account for variable follow-up, our hypothetical researcher is systematically introducing bias into their sample and may produce a misleading count of the number of stopped trials. To correct for this, the overall status of every trial in the sample must be assessed at the same follow-up time; trials where the requisite follow-up time has not yet passed must be excluded from analysis.

The script in R Code Box 3 will download historical clinical trial data for the trial numbers specified in the CSV downloaded from a search using the web front-end of ClinicalTrials.gov. Trials with less than five years of follow-up will be removed from the sample, and the script

```
## R Code Box 3

library(tidyverse)
library(cthist)
library(lubridate)

## To reproduce these methods, conduct a search on the
## ClinicalTrials.gov web front-end and download the result as a
## comma-separated value (CSV) file named `SearchResults.csv`. Save
## this file to your R working directory. This file provides `cthist`
## with the list of NCT numbers for which to download historical trial
## registry data.
trials <- readr::read_csv("SearchResults.csv") %>%
    select(`NCT Number`) %>%
    pull(`NCT Number`)

## Download historical clinical trial data for all NCT numbers in the
## CSV downloaded from ClinicalTrials.gov
clinicaltrials_gov_download(trials, "historical_versions_3.csv")

## Read historical versions from the data in the CSV generated above
hv <- read_csv("historical_versions_3.csv")

## Define follow-up time as 5 years
followup <- lubridate::years(5)

## Make a new column for the first version date for each trial and
## filter for the version of the clinical trial registry that was
## active at 5 years after that date
overall_statuses <- hv %>%
    group_by(nctid) %>%
    mutate(first_version_date = min(version_date)) %>%
    filter(
        as.Date(version_date) <=
        as.Date(first_version_date) + followup
    ) %>%
    slice_tail() %>%
    ungroup() %>%
    filter(
        first_version_date <=
        Sys.Date() - followup
    ) %>%
    select(
        nctid, first_version_date, version_date, overall_status
    ) %>%
    mutate(
        stopped = overall_status %in%
```

```
            c(
                "Suspended",
                "Terminated",
                "Withdrawn"
            )
    )

## Count result
overall_statuses %>%
    count(stopped)

## Write result as a CSV to disk
overall_statuses %>%
    write_csv("overall_statuses.csv")
```

will write a CSV file to disk that includes the overall status of each eligible trial at five years after the initial registration.

Among the 93 phase 3 glioblastoma trials on ClinicalTrials.gov as of 2022-01-05, 69 have 5 years of follow-up, and 10 of those (14.4%) had a status of "Terminated", "Suspended" or "Withdrawn" at 5 years. Among the 6 trials that have overall statuses of "Terminated", "Suspended" or "Withdrawn" on the most recent version on ClinicalTrials.gov but not at 5-years of follow-up, 4 were originally registered less than 5 years ago, and the remaining 2 changed their status to "Terminated", "Suspended" or "Withdrawn" after the 5-year mark.

## Discussion

The purpose of prospective registration of clinical trials is partly vitiated if there is no efficient means to access historical clinical trial registry data. The responsible party for a clinical trial can effectively "bury" changes to a registry entry in its history if there is no feasible way to access history changes. Peer-review of individual clinical trial journal publications does not provide sufficient scrutiny to ensure that the publication is not misleading or biased due to outcome switching, etc. as trial registration records are not always thoroughly checked for accuracy [25]. While the disclosure of changes to individual trial registry entries through the registry website provides some level of openness, this only allows researchers to find registry entry changes if they already know where to look, and limits the feasibility or reproducibility of certain kinds of research.

The cthist package provides functions that allow for efficient mass-downloading and processing of historical clinical trial registration data from ClinicalTrials.gov and DRKS.de. This makes certain kinds of meta-research feasible and provides a means to correct common errors in data collection and analysis, such as overlooking variable follow-up. This package also increases the reproducibility of previously completed analyses of clinical trial registry data. Among analyses of clinical trial registry entries, it is common practice to report the date that the clinical trial registry was searched, as the database's contents change frequently. Without a way to select data from a specific date as provided by cthist, reproducing this kind of research is difficult or impossible.

## Limitations

The *R* package described here is a web-scraper that provides a means for retrieving historical clinical trial data that is not easily available without extensive manual work. Web-scraping a clinical trial registry differs from accessing it through an API in that an API is designed to be queried repeatedly and receive a large volume of requests from automated programmes, whereas a web-scraping tool has opportunistically repurposed a resource that was originally designed only to be accessed by individuals manually through a web browser. This means that ClinicalTrials.gov or DRKS may change their websites at any time in ways that alter CSS selectors on which this web-scraper depends, for example [26]. An effort has been made in the design of this tool to respect server requests regarding the volume and frequency of queries by implementing server calls using the `polite` *R* package [20], however there is a risk that ClinicalTrials.gov or DRKS may implement changes at any time that intentionally or unintentionally break the functionality of this package.

Further, not all data points that are available in ClinicalTrials.gov or DRKS.de are collected by `cthist`, although the package is easily extensible to collect anything reported on the historical version page.

## Future directions

The WHO registry network lists 17 primary clinical trial registries other than ClinicalTrials.gov (including DRKS.de) [27]. Future versions of this package may include functions for downloading historical clinical trial registry data from other clinical trial registries that provide access to historical versions through their website. Future versions may also include functions for downloading additional data points from ClinicalTrials.gov and DRKS.de.

This *R* package may also be integrated into an automated tool to generate reports for reviewers of clinical trials in partnership with journals who publish clinical trials results. This report could include key information on a clinical trial based on data extracted by `cthist` to assist in their reviews. A prototype of such an application is available [28].

It is also my hope that the existence of this *R* package may draw attention to its necessity by those who make design decisions for clinical trial registries, and implement means for mass-downloading historical clinical trial data for analysis that do not require the use of this *R* package.

## Acknowledgments

This work was informed and motivated by clinical trial research projects involving Delwen Franzen, Martin Haslberger, Martin Holst, Nora Hutchison, Katarzyna Klas, Maia Salholz-Hillel, Daniel Strech and Marcin Waligora. We acknowledge financial support from the Open Access Publication Fund of Charité –Universitätsmedizin Berlin and the German Research Foundation (DFG).

## Author Contributions

**Conceptualization:** Benjamin Gregory Carlisle.

**Data curation:** Benjamin Gregory Carlisle.

**Formal analysis:** Benjamin Gregory Carlisle.

**Investigation:** Benjamin Gregory Carlisle.

**Methodology:** Benjamin Gregory Carlisle.

**Software:** Benjamin Gregory Carlisle.

**Writing – original draft:** Benjamin Gregory Carlisle.

**Writing – review & editing:** Benjamin Gregory Carlisle.

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
