## [Decision Letter · Decision Letter 0]

14 Mar 2022

PONE-D-22-03050Analysis of clinical trial registry entry histories using the novel R package cthistPLOS ONE

Dear Dr. Carlisle,

Thank you for submitting your manuscript to PLOS ONE. After careful consideration, we feel that it has merit but does not fully meet PLOS ONE’s publication criteria as it currently stands. Therefore, we invite you to submit a revised version of the manuscript that addresses the points raised during the review process.

**First of all, I would like to thank all the 4 reviewers**. They were fast in answering and provided a lot of minor but very constructive points. Please address all these points very carefully. I also want to say that based on my own review, I think that the paper is sound and that it will be very helpful for meta-researchers involved in this field as it details a very important R library. I will be therefore very pleased to see a revised manuscript. I have 2 additional comments :  - Please add a few words about the main limitations of the library in the abstract ; - Please detail with more details any plan you have to update the library in the future ;  My decision is major revision 1/ because of the large number of minor revisions requested by the reviewers and 2/ because after this first round of peer review, I will send it again to the reviewers. In order to be transparent, please note that I was aware that one of the 4 reviewer had a potential (non-financial) conflict of interest. However, I was pretty sure that his input would be very helpful in order to strengthening the manuscript and I have asked him to review it. Please note that I did not take his "minor revision" suggestion into account when making my decision. For this reason, I invited 4 reviewers and only based my decision on the 3 remaining reviewers. However, as you will see, inter-reviewers agreement was really good.

We look forward to receiving your revised manuscript.

Kind regards,

Florian Naudet, M.D., M.P.H., Ph.D.

Academic Editor

PLOS ONE

Journal Requirements:

2. We noted in your submission details that a portion of your manuscript may have been presented or published elsewhere. [This publication is available as a preprint: https://www.medrxiv.org/content/10.1101/2022.01.20.22269538v1] Please clarify whether this [conference proceeding or publication] was peer-reviewed and formally published. If this work was previously peer-reviewed and published, in the cover letter please provide the reason that this work does not constitute dual publication and should be included in the current manuscript.

Reviewers' comments:

Reviewer's Responses to Questions

**Comments to the Author**

1. Is the manuscript technically sound, and do the data support the conclusions?

Reviewer #1: Yes

Reviewer #2: Yes

Reviewer #3: Yes

Reviewer #4: Yes

2. Has the statistical analysis been performed appropriately and rigorously? 

Reviewer #1: Yes

Reviewer #2: N/A

Reviewer #3: N/A

Reviewer #4: N/A

3. Have the authors made all data underlying the findings in their manuscript fully available?

Reviewer #1: Yes

Reviewer #2: Yes

Reviewer #3: Yes

Reviewer #4: Yes

4. Is the manuscript presented in an intelligible fashion and written in standard English?

Reviewer #1: Yes

Reviewer #2: Yes

Reviewer #3: Yes

Reviewer #4: Yes

5. Review Comments to the Author

Reviewer #1: This is an interesting paper that presents an R package that performs web scraping to get historical data on clinical trials from two registries : clinicaltrials.gov and DRKS. The code source of the package is open source on GitHub.

Some parts deserve a more elaborate description:

- the clinical trials API is not mentioned at all: from what I understand, the API does not provide the historical data and web crawling is the only way to get it, but still this should be clarified in the paper. The choice to use web crawling techniques has to be justified as using an API would be a much simpler way to proceed if it would be possible

- web scraping is a very useful technique but should be done in a "polite" way, to make sure not to disturb the underlying service / website. The way the package is used depends on the final user of course but this issue could be brought to the attention of the reader

- The 3 cases study are very interesting applications of the package. But only the third comes with results explaining the benefits of the historical analysis. At least some light results could be provided also for the first two cases study.

- in Discussion / Future directions: clinicaltrials.gov and DRKS do provide access to webpages describing the historical changes : that may not be the case for other registries so this method may not be extended as such to any registry.

Reviewer #2: First of all, I would like to thank the editor for the invitation to review this paper and draw my attention to this interesting R package. I think the package is a valuable addition to the R package ecosystem, as there are only very few, if any, packages that allow downloading clinical trial data (some older packages have been archived). Additionally, as the author has correctly pointed out, the usual methods for downloading trial data (e.g., the AACT) do not include the history of changes to a registry entry.

The package itself could be described as rather minimalistic but worked well during my testing, and the paper describes the included functions quite well. The case studies illustrate briefly but convincingly how the package enables analyses that would otherwise require considerable manual work.

I could successfully run the code for all case studies and briefly tested the Shiny app, which also worked well. However, after reviewing the code and the article, I have some comments.

Comments regarding the package and code:

Comment: When I first tried the clinicaltrials_gov_dates function, it returned some correct dates but many NAs. It turned out that this was due to my language setting so that only some month names could be read in. I could avoid this error by temporarily changing the locale:

lct <- Sys.getlocale("LC_TIME") # backup

Sys.setlocale("LC_TIME", "C")

Maybe the function could do this automatically (and restore the original locale afterwards)? This will otherwise lead to frequent problems.

Comment: The function should probably also warn if it returns NAs.

Comment: I realized the above when running the code line-by-line. I noticed that the line 'format("%Y-%m-%d")' in clinicaltrials_gov_dates seems to be superfluous, but maybe it is meant as a 'double-check' to make sure the dates are formatted correctly.

Comment: This comment is about both the article and the code. I think it is important to correctly name the returned value of clinicaltrials_gov_version (and the respective function for DRKS). The article often mentions 'lists' or 'data frames' (e.g., line 160), and the help for clinicaltrials_gov_version defines the returned value as a list, although it is actually a character vector.

Comment: In connection with the previous comment, I believe that it might indeed be better to let that function return a list instead of a character vector. First, lists are printed more nicely in the R console, and second, if the elements of the list were named appropriately (with 'outcomes', 'criteria', and so on), we could use $ to subset the returned list. Does returning the result as a character vector have any advantages?

Comment: I have tested the download function for ClinicalTrials.gov with 100 trials for a certain query and also the 93 trials on glioblastoma, as demonstrated in case study 1. However, all values for gender_based were missing. Is this plausible?

Comment: It is good that the author has written unit tests for cthist. However, I think the tests are currently quite basic and could easily test the output more thoroughly. For example, test-clinicaltrials_gov_version.R could test for missing values and correct date format (or even some specific values), instead of only testing the length of the vector.

Comment: It would be nice if ?cthist had a help page with a short overview of the package.

Comment: ClinicalTrials.gov has announced an update to the website. Do you know if this will interfere with the scraper's functionality? Luckily, the beta version of the history page looks identical to the current version at first glance.

Comment: It seems a bit unusual to me that some of the functions, such as the *_dates functions) return "Error" and "Warning" as character vectors upon failure and print the warnings and errors using message(). I assume the intention is not to interrupt longer jobs if only very few tasks fail, but again, this is somewhat unusual because a character vector is the returned data format upon success. Typically, the warning() and stop() functions would be used here. Maybe the functions could have a parameter to let the user choose the behavior? I'm not sure, to be honest.

Comment: The returned data class from the *_dates functions is a character vector, but the help states that it is a date vector (which would be better, in my opinion).

Comment: Are you aware of any limits on automatic requests to ClinicalTrials.gov or DRKS.de? Some sites block web scrapers after a certain amount of requests. I don't think this is the case with these two sites, but it would be good to confirm that. Some users might download a very large number of trials.

Comments regarding the article:

Comment: The case studies are short, but convincing. It would be good to show or mention at least some of the results for every case study instead of leaving this up to the reader.

Comment: Maybe it should be clearly mentioned that 'version 1' is the oldest version of a trial entry. It would be good also to say this on the R help pages.

Comment: The package focuses on the history of registry entries, but I could imagine it being used for the general downloading of current entries. Since the rclinicaltrials package has been archived, there is no package with this functionality on CRAN, as far as I know. It would be helpful to briefly describe how to download only the most current entries. A 'trick' for doing so using slice_tail is included in one of the code examples but could be missed by readers easily.

Comment: Line 114 states "enrolment type (“Anticipated” or “Actual”), which are split into separate columns by the clinicaltrials_gov_download() function". It seems that actually 'Anticipated' or 'Actual' are included after the enrolment value in element 2 and additionally in element 5 of the returned vector. I assume element 2 should be only the numerical value?

Comment: Is there a way to get the text 'cleaned' from line break code etc.? Could this be done by the package or are there other R packages that are better suited for that? Most users will probably want to clean these entries. A solution could be briefly mentioned.

Comment: I could successfully convert the JSON values that are returned by some of the functions. However, the article mentions in several places that, e.g., the outcome measures are 'data frames', although it seems to me R will convert the JSON to lists (when using rjson). Is this correct? I think returning an actual data.frame as a list element would actually be a more user-friendly solution than the current one, but I assume this was done to keep the output identical to the JSON entries in the CSV generated by the download function. Anyway, the article should be clear that the package does not return data frames or how the returned values could be converted to data frames.

Comment: Line 147: Maybe mention that 'Percentage change' refers to the percentage change of the length of the recruitment period.

Comment: Some references are missing DOIs. I also think that the citations of R packages should contain a link to CRAN, if applicable.

Reviewer #3: The author present a new package for the R statistical programming language, aimed as streamlining web scraping of two major clinical trial registries : ClinicalTrials.gov and DRKS.de. This package help retrieving change history of applications, a function which is not present in these registries API. It hints at an inclusion in reviewing of clinical trial, which would indeed be a useful complement to reported protocol changes.

In the discussion, the author press that the existence of this tool may reveal it's additional value, which is an important point.

This package is self-contained and usable without disturbance in multiple R paradigms. A version which do not request disk access and is able to operate fully on RAM may be prefered but is functional as is.

A) Regarding the article :

A.1) Abstract : I may advise to change "mass downloading" to "scraping", which is more precise while hinting at volume tolerance of the method. I am also surprised by the expression "the enterprise of human research", which I am not sure is widely used in english. As I am not a native speaker, this is just a question.

A.2) In the introduction, the author write "Certain research questions [...] are not possible or not feasible to address without a means of accessing and analyzing the history of changes for an entire cohort of clinical trials in a systematic way." I would be pleased to be presented with one or two examples.

A.3) Again in the introduction, the author write "There is no API access to historical clinical trial registry data for

ClinicalTrials.gov or DRKS.de." As there is in fact an API for both of these registries, I would have like a more precise description of the limitation of theses API and the need of this webscraping package.

When visiting ClinicalTrials.gov API webpage, it call up the Clinical Trials Transformation Initiative (CTTI)'s Database for Aggregate Analysis of ClinicalTrials.gov (AACT). Maybe add a sentence to discuss the objective of this base and the limitations motivating the creation of the cthist package ?

A.4) In the method part, the details concerning the *_version functions seems shared between both, and the paragraphs have many repetition making reading it slightly laborious. Does the integration of more technical parts in the table 1 make in fact for a more or less concise and reader-friendly presentation ?

Complementary, regexes are mostly simple, but as a "write only" language it may be best to systematically add a translation in plain english. As there is mostly 3 different regexes, they may be substituted in the text by the plain english translation.

A.5) Again in the method part, I would like to have more precision on the two following affirmations :

"These functions also implement automated error-checking on completion, in order to ensure the accurate retrieval of large sets of clinical trial registry entry histories. If any version of the clinical trials to be downloaded returned an error, these functions return FALSE, otherwise they return TRUE." and "These functions also implement automated checking for already-downloaded data when starting, to allow for re-starting partially completed downloads."

Which further details on what type errors are checked and to what extend : do the function only return the website error ? Are errors sources furthermore explored ? etc.

For the second and from what I read in the code, it only check if a data is already present and complete, accelerating complementary download more than explicitly "allowing" it ?

A.6) For Case study 3, even as it is just an example, why not keep trials which and describing status completion as completed, terminated or censored at extraction date (and statut regarding the exceeding of forecasting end date) ?

A.7) In the discussion part : As this package is also aimed at researchers not accustomed with web scraping, I would advise to add one or two paragraphs introducing to the basics of web scraping : differences between using an API and scraping a page, the risk of brutal changes and end of service, websites and other users respect regarding requests volume and frequency, risk of IP ban and other countermeasures, etc.

B) Regarding the code examples :

B.1) For all code examples, adding a (succinct) comment on each line code may be useful for readers not accustomed with the pipe workflow and tidyverse-specific function.

B.2) This phrase is really hard to read and understand until later in the code, and may benefit of a refactoring centering on objectives in the style of "extract initial definitions of start and completion date"

"## Define the start date at launch and completion date at launch to be

## the start date and completion date reported on the first version of

## the trial where that version's overall status is listed as

## "Recruiting" "

B.3) In the pipe `readr::read_csv("SearchResults.csv") %>% select(`NCT Number`) %>% pull() `, using `pull()` without argument is at risk. `readr::read_csv("SearchResults.csv") %>% pull(`NCT Number`)` should be more reliable.

B.4) The definition of "1 year" beeing variable in number of days and `followup <- 365` carrying a risk of error, it may be beneficially replaced by `dmy("21-01-2021") + lubridate::years(1)`, taking care of leap year.

B.5) In code box 2, the use of an external rle object may not be apparent for readers. The author may add an reason for the use of run length, or maybe go for a whole-in-one approch as in :

```

hv %>%

mutate(outcome_run =

paste(drksid, primary_outcomes, secondary_outcomes) %>%

{. != lag(.)} %>% coalesce(TRUE) %>% cumsum()

) %>%

group_by(outcome_run) %>%

slice_head() %>%

ungroup()

```

B.6) In the code box 3, it is possible to additionally reduce redundancy by using `stopped = overall_status %in% c("Suspended", "Terminated", "Withdrawn")` in line 491.

C) Regarding the package in itself :

As a first point, I wish to emphasize the global quality of the package design and the use of dependency for import of other packages. The following comments may in no point influence the publication decision as they relate to the package in itself and not the article presenting it. The three major limitation that I see as of now are :

C.1) The lack of defensive programming in the current state, with no testing of function argument and thus no meaningful error code and no catch of error before handling to the registry website. As NCT numbers format is strictly defined, adding verification that a character vector of length one and corresponding to a defined regex may be a critical add, ensuring that calls like `clinicaltrials_gov_version("bad_id", "2,8")` or `drks_de_version("DRKS0000000000005219", glm(Species ~ Sepal.Length, data = iris))` have informative errors.

C.2) For all extracted dates, filling in missing data may be misleading in creating false precision. I would advise to either : issue a warning or, preferably, keep dates as plain text and let the user deal with imprecision in day number in the way that is more effective in the specific study context (first day, last day, interval, discard, etc.)

C.3) By the actual design, the *_download() functions request a writing disk access, which : 1) may be difficult in some secure analysis environnement, and 2) is a deviation on the general R objective of "no side effect". For the cleanest effect, it may be preferable to rewrite the *_download() to only batch call the *_version() functions and let the user choose how to keep its data and control it's storage. A simple alternative may be to change `output_filename` default value to `output_filename = tempfile()`, storing values in RAM, and make the function return the data frame (invisibly if needed) in place of an indication of success or failure.

Additional points that may streamline the package starting and usage :

C.4) When screening the code, the *_version() and *_dates() function seems to be backend for by *_download() function. If they are not intended to have separated use, they may be keeped as internal function and only *_download() function may be exported : https://r-pkgs.org/namespace.html . This way there will also be less function help-pages to write.

In addition, *_version() function are the ones actually downloading the informations and return it in R and *_download() function map *_version() functions and write it to the disc, which is slightly misleading.

C.5) If *_version() function are keeped exposed, adding a default version number value may be useful, for example `versionno = 0`, and defensive programming added to it too.

C.6) The help files may be slightly thin, and adding one of the article example may be of great benefit, especially the explanation of how to get a csv of search result and not having to manually copy the individual NCT numbers. If it is simple enough, do the author plan to add a function taking a research string as an argument and returning the NCT numbers ?

C.7) To avoid IP ban or accidental DOS attack, is there a timer between two requests by *_version() functions ? Adding one, setting it as arbitrarily large (for example 1 second) and presenting it as an argument would be useful at the same time as warning against error and as teaching.

Reviewer #4: Many thanks for the opportunity to review this piece explaining a new R package to examine historic trial information at ClinicalTrials.gov and the DRKS.

The purpose of this package is well described and justified. Working with historic registry data is something I have had to do, and write ad hoc code to handle (though in Python for my cases), for a number of my projects. Creating a package that handles at least some of these aspects of this type of task is certainly welcome and valuable. I have separated my comments into three parts covering my use of the package, the code, and the text of the manuscript. I note some minor points all around.

Comments on Usage:

While R is not my primary language I was able to install the package from CRAN and work through the various functions. For the author’s reference, I was working in R v4.0.5 within RStudio v1.4.1106 on a Mac running 10.15.17.

I cross checked outputs with a few sample trials from each registry and found no discrepancies. I admit, my lack of experience in R likely limited my ability to play around with the data in more depth, but the outputs are all very simple. It’s either a vector of dates, or a character vector containing strings of the variables in the relevant fields that can be easily outputted to CSV. The more complex fields are organized and tagged logically (in a way similar to a Dictionary in Python).

One note is that when I ran the example function in the documentation for downloading multiple ClinicalTrials.gov records: `clinicaltrials_gov_download(c("NCT02110043", "NCT03281616"), "test_review.csv")` it worked just fine to actually download the data and output the file but I got a parsing error on each row in which it expected 18 columns and got 19. The output is copied below. This might be unavoidable for some technical reason, but if that is the case perhaps note this in the documentation so that people aren’t thrown off by the errors during use. The DRKS example did not do this.

clinicaltrials_gov_download(c("NCT02110043", "NCT03281616"), "test_review.csv")

022-03-09 12:49:00 NCT02110043 processed (8 versions, 50%)

2022-03-09 12:49:06 NCT03281616 processed (2 versions, 100%)

Warning: 10 parsing failures.

row col expected actual file

1 -- 18 columns 19 columns 'test_review.csv'

2 -- 18 columns 19 columns 'test_review.csv'

3 -- 18 columns 19 columns 'test_review.csv'

4 -- 18 columns 19 columns 'test_review.csv'

5 -- 18 columns 19 columns 'test_review.csv'

... ... .......... .......... .................

See problems(...) for more details.

[1] TRUE

Warning messages:

1: Unnamed `col_types` should have the same length as `col_names`. Using smaller of the two.

2: Unnamed `col_types` should have the same length as `col_names`. Using smaller of the two.

Otherwise, I think this is all relatively straightforward and anyone with familiarity working in R should be able to get from these simple functions to workable data relatively easily.

Comments on the Code:

Once again, slightly limited by my knowledge of R here, but I did take a look over the code for the package on GitHub. Since, as the author notes, there is no API for either registry that can be used for historic data, this package essentially runs a scraper in the background to gather the relevant data from the history pages. This is noted in the limitations of the paper, but you might also want to include in your documentation so people can understand that very large queries might cause issues at the host registry.

While I’m not in a position to comment on any by-line bugs or enhancements, given my own experience scraping historic registry data, I was able to follow how each function broadly worked, including how the `download` functions call the other functions to operate and it all made sense from an implementation perspective. I especially like the feature to re-check for partial downloads to not waste time and pings of the registry if a scrape is interrupted.

I was able to get all the code examples to run and produce expected results on my own.

Comments on the Text:

Since you have a table that matches each extracted field to it’s CSS selector, I think you can simply point to that rather than also write it out in words from lines 108-163 and 175-206. If you want to keep details of any context you feel is necessary about the extractions in prose, that is fine but there’s no need to keep repeatedly referencing the CSS selectors and HTML elements used when the Table does that much more efficiently for anyone who is interested. Otherwise it really hinders the readability of that section.

6. PLOS authors have the option to publish the peer review history of their article (what does this mean?). If published, this will include your full peer review and any attached files.

Reviewer #1: **Yes: **Eric Jeangirard

Reviewer #2: No

Reviewer #3: **Yes: **Scanff, Alexandre

Reviewer #4: **Yes: **Nicholas J. DeVito

---

## [Author Response · Author response to Decision Letter 0]

27 Apr 2022

Please see attached file, `Response to Reviewers.docx`

---

## [Decision Letter · Decision Letter 1]

7 Jun 2022

PONE-D-22-03050R1Analysis of clinical trial registry entry histories using the novel R package cthistPLOS ONE

Dear Dr. Carlisle,

Thank you for submitting your manuscript to PLOS ONE. After careful consideration, we feel that it has merit but does not fully meet PLOS ONE’s publication criteria as it currently stands. Therefore, we invite you to submit a revised version of the manuscript that addresses the points raised during the review process.

I would like to thank the 4 reviewers. I have no additional comment. 

We look forward to receiving your revised manuscript.

Kind regards,

Florian Naudet, M.D., M.P.H., Ph.D.

Academic Editor

PLOS ONE

Journal Requirements:

Reviewers' comments:

Reviewer's Responses to Questions

**Comments to the Author**

1. If the authors have adequately addressed your comments raised in a previous round of review and you feel that this manuscript is now acceptable for publication, you may indicate that here to bypass the “Comments to the Author” section, enter your conflict of interest statement in the “Confidential to Editor” section, and submit your "Accept" recommendation.

Reviewer #1: All comments have been addressed

Reviewer #2: All comments have been addressed

Reviewer #3: (No Response)

Reviewer #4: All comments have been addressed

2. Is the manuscript technically sound, and do the data support the conclusions?

Reviewer #1: Yes

Reviewer #2: Yes

Reviewer #3: Yes

Reviewer #4: Yes

3. Has the statistical analysis been performed appropriately and rigorously? 

Reviewer #1: Yes

Reviewer #2: N/A

Reviewer #3: N/A

Reviewer #4: N/A

4. Have the authors made all data underlying the findings in their manuscript fully available?

Reviewer #1: Yes

Reviewer #2: Yes

Reviewer #3: Yes

Reviewer #4: Yes

5. Is the manuscript presented in an intelligible fashion and written in standard English?

Reviewer #1: Yes

Reviewer #2: Yes

Reviewer #3: Yes

Reviewer #4: Yes

6. Review Comments to the Author

Reviewer #1: (No Response)

Reviewer #2: The author has made extensive changes to both the paper and the package. Both are in a much-improved state compared to the previous submission. My comments have been addressed sufficiently, and I would like to thank the author for his detailed responses. In my opinion, the article is now acceptable for publication.

Reviewer #3: I would like to thanks the author for his responses and clarifications. The context and the workflow benefit from more in-depth description and the integration of the "polite" package is a great quality improvement of the cthist package.

I now only have two tiny comments :

- In response for the inquiry about DRKS API : I had found this aggregator service via clarivate https://www.cortellislabs.com/page/?api=api-CLI . It may not be in fact sufficient for the intended use, but may be worth mentioning.

- The code comment referred in B.2) "## Define the start date at launch [...]" is mentioned as edited but have no tracked change. Have one of the intended change been slipped away between versions ?

Thanks again for this new tool in the R environment and the opportunity to review this package

Reviewer #4: (No Response)

7. PLOS authors have the option to publish the peer review history of their article (what does this mean?). If published, this will include your full peer review and any attached files.

Reviewer #1: **Yes: **Eric JEANGIRARD

Reviewer #2: No

Reviewer #3: **Yes: **Alexandre Scanff

Reviewer #4: No

---

## [Author Response · Author response to Decision Letter 1]

17 Jun 2022

Reviewer #3:

I would like to thanks the author for his responses and clarifications. The context and the workflow benefit from more in-depth description and the integration of the "polite" package is a great quality improvement of the cthist package.

I now only have two tiny comments :

- In response for the inquiry about DRKS API : I had found this aggregator service via clarivate https://www.cortellislabs.com/page/?api=api-CLI . It may not be in fact sufficient for the intended use, but may be worth mentioning.

 • A reference to the 3rd-party API has been added to the manuscript. See lines 51-54.

- The code comment referred in B.2) "## Define the start date at launch [...]" is mentioned as edited but have no tracked change. Have one of the intended change been slipped away between versions ?

 • A comment beginning “## Define the start date at launch” appears twice in the manuscript. In both cases, a clarifying paragraph has been added afterward. See lines 354-369 and 390-402.

Thanks again for this new tool in the R environment and the opportunity to review this package

 • Thank you for your review!

---

## [Editor Report · Decision Letter 2]

20 Jun 2022

Analysis of clinical trial registry entry histories using the novel R package cthist

PONE-D-22-03050R2

Dear Dr. Carlisle,

We’re pleased to inform you that your manuscript has been judged scientifically suitable for publication and will be formally accepted for publication once it meets all outstanding technical requirements.

Kind regards,

Florian Naudet, M.D., M.P.H., Ph.D.

Academic Editor

PLOS ONE
---

## [Editor Report · Acceptance letter]

23 Jun 2022

PONE-D-22-03050R2 

Analysis of clinical trial registry entry histories using the novel R package cthist 

Dear Dr. Carlisle:

I'm pleased to inform you that your manuscript has been deemed suitable for publication in PLOS ONE. Congratulations! Your manuscript is now with our production department. 

Kind regards, 

on behalf of

Pr. Florian Naudet 

Academic Editor

PLOS ONE